# A Study on the Causal Process of Virtual Reality Tourism and Its Attributes in Terms of Their Effects on Subjective Well-Being during COVID-19

**DOI:** 10.3390/ijerph18031019

**Published:** 2021-01-24

**Authors:** Yu Li, HakJun Song, Rui Guo

**Affiliations:** 1School of Management, Bohai University, Jinzhou 121007, China; brianlee@bhu.edu.cn; 2College of Tourism and Fashion, Pai Chai University, Daejeon 35345, Korea; guoruippx@gmail.com

**Keywords:** virtual reality tourism, subjective well-being, peripheral attribute, core attribute, pivotal attribute, presence, perceived value, satisfaction, Quanjingke

## Abstract

In the landscape of Covid-19 pandemic, people’s well-being, to some extent, can be affected through virtual reality tourism because it has the opportunity to enhance their level of well-being and destination recovery. To verify this empirically an investigation was conducted among people who used Quanjingke, the largest tourism-related virtual reality website in China, during the pandemic. The specific the aim of this paper is to prove the effectiveness of virtual reality tourism in promoting people’s subjective well-being. Hence, an integrated model with the constructs of peripheral attribute, core attribute and pivotal attribute, presence, perceived value, satisfaction, and subjective well-being was proposed and tested. The results indicate that attributes of virtual reality tourism have a positive effect on presence during virtual reality experiences, which positively influences perceived value. The values of virtual reality tourism perceived by tourists result in their satisfaction. Eventually, it was found that tourists’ subjective well-being is improved due to their satisfaction with virtual reality tourism. Practical suggestions are also provided based on the findings.

## 1. Introduction

From the beginning of 2020 up to now, most of the world has been struggling with the COVID-19 pandemic. Rigorous restrictions, like entry bans and quarantines, and preventive measures are imposed throughout the world to halt the spread of the coronavirus, causing a downturn in economic activity and sapping the economic momentum of countries. There is no exception for the tourism industry with regard to such a recession. The COVID-19 pandemic has caused a 70% fall in international tourist arrivals (overnight visitors) during first eight months of 2020 compared to the same period of the previous year. Particularly, international arrivals declined 81% in July and 79% in August, which would usually be the peak season of the year, the latest data from the World Tourism Organization (UNWTO) indicates [1].

Considering the quality of life and residents’ well-being, our lives have been dramatically changed by the pandemic. Waves of strict lockdowns are “inevitable” and social distancing is required by governments and organizations to prevent virus transmission, which, to some extent, induce negative effects on the mental health and well-being of residents and the effects will extend beyond those who have been directly affected by the virus [2]. Ma and Yang found that the onset of the coronavirus epidemic led to a 74% drop in overall emotional well-being which is an important constituent of subjective well-being (SWB) [3]. Given the importance of SWB to residents’ lives, it is time to take reasonable precautions to help us bring health and well-being to the forefront. Recommendations that could contribute to an alteration in residents’ well-being, including the promotion of physical activity and sleep hygiene etc. are contained in the relevant literature [4]. The improvement of the well-being of humankind has been an object of many public policies [5]. On the individual level, experiences during travel and satisfaction with travel contribute to improving well-being [5]. Given the risk of increasing the opportunities for spreading the virus and getting infected, it would be better to find an alternative way to travel without physical movement when travels are postponed on a large scale and staying at home is advocated.

In the current world, which has been filled with artificial intelligence, we are becoming increasingly reliant on technology. For example, virtual reality (VR) is expected to be one of the significant technology products in the tourism industry. By providing accessible tourism for all and enhancing tourist experience, immersion, and visualization, VR may be an invaluable resource for transforming traditional tourism with intangible and experiential essence into a product [6,7]. Therefore, it seems that VR tourism has the potential to provide residents with the access to experience tourist sites in VR and can play a vital role in improving residents’ well-being [8]. 

The subject of this study, Quanjingke (QJK) provides 360-degree and ultra-high-definition panoramic images and guided tours and its language is Chinese only. According to the latest report from one of the most authoritative websites in China, QJK is the largest tourism-related VR website in China and it has around 1.5 billion active users and provides over 60,000 panoramic pictures and VR videos all over the whole country [9]. The large amounts of users of QJK and rich VR resources make it reliable for collecting meaningful data for the current study. At the early stage of the pandemic, it is noteworthy that an article page published on QJK’s official account of WeChat induced the number of 67 million page views, leading to over 40 million unique visitors and 150 million page views for its website and its popular app, “Beautiful China” [9]. Many Chinese accepted the new idea of “take it home”, which means to encourage potential tourists to “carry” tourist resorts to their home by using laptops and smart phones via which the interesting photos or videos and intelligent tour guides can be attained during the pandemic. With rich VR tourism resources and large amounts users, QJK enables us to access high quality data for empirical research related to VR tourism in the context of the pandemic.

The subjective evaluation of well-being is frequently referred to as SWB, and the subjective appraisal of well-being is the measure of well-being achieved when answering questions, which means that SWB can be measured by questionnaire in a self-reported way [10]. The adoption of VR associated with enhancing subject well-being has been examined in various contexts. For example, Li et al. investigated the effectiveness of using virtual reality computer games in promoting the subjective well-being of children with cancer [11]. In spite of this, exploring the role of VR tourism on enhancing residents’ subjective well-being remains in its infancy, as an integrated model of VR tourism has not been established [12]. To be concrete, researchers emphasize certain facets of VR tourist behavior (e.g., perception of authenticity and attitude) for the purpose of tourism marketing [12]. In addition, a handful of the literature employing theoretically integrated model remains on too broad scope rather than concentrating on a specific VR tourism product, leading to potentially diverse conclusions. Focusing on the VR tourists of QJK, we strive to fill the aforementioned research gaps by establishing an integrated model. In this study, structural equation modeling (SEM) was employed to test the proposed model and hence the relationship between VR tourism and residents’ subjective well-being was explicated clearly on both theoretical and empirical level.

Incorporating the constructs of PCP attributes, presence, perceived values, and satisfaction into an integrated model, the current study aims to explore how the VR tourism promote residents’ subjective well-being by delving into its mechanisms. The model incorporates constructs frequently used in the consumer behavior domain, encompassing PCP attributes (i.e., peripheral attribute, core attribute and pivotal attribute), perceived value (i.e., functional value and emotional value), and satisfaction, with the construct unique to VR tourism research (i.e., presence). Consequently, the theoretical and practical implications are summarized based on the results of empirical analysis. Thus, the policy makers, residents, tourist destinations and VR tourism operators will all benefit from the research findings. 

## 2. Literature Review

### 2.1. Attributes of VR Tourism and PCP Model

Through the era of PC, realistic online (including mobile) content that stimulates the five senses through VR (virtual reality) and AR (augmented reality) is growing. Virtual reality refers to a system that creates a three-dimensional visual and auditory experience in real time and expresses an object in a simulated form [13]. The concept of virtual reality began to be used as a theoretical approach in the field of HCI (Hyper Converged Infrastructure) in the mid-1970s, but it began to be actively used in the 1980s. Various studies and attempts have been made in the field of virtual reality [14,15]. In addition, as related contents increase, various distributions are being made in connection with culture, art, sports, and tourism. In the future, more fields using virtual reality technology such as games, education, medical care, manufacturing, and e-commerce are expected to increase. Burdea and Coiffet defined virtual reality as an interface between humans and computers that enables users to immerse themselves and interact in real time [15]. As one of the fields of virtual reality application, VR tourism (with the synonyms “virtual tour”, “panoramic tour” etc.) provides VR tourists with the online service to experience traveling in virtual environments by creating multimedia elements and simulating real tourist sites and unreal situations [8].

Just as e-service enterprises in the travel industry offer various online services (travel planning, hotel reservations, and rental car services), VR tourism operators provide various types of virtual services related to destination experiences [12]. Therefore, it is crucially important to grasp the general properties of VR tourism and to measure VR tourists’ evaluation of the service performance of VR tourism in order to explain the mechanism by which VR tourists improve subjective well-being through VR travel.

Scholars have proposed the various attributes of e-service quality based on their context. Among various e-service quality-related studies, several studies that are related to VR tourism are as follows. Argyriou et al. proposed five primary attributes of VR tourism (“narrative”, “virtual scenes”, “actor role”, “navigation”, “gamified”) that are important to VR tourism quality [16]. Chiao et al. identified a virtual reality tour-guiding platform consisting of “itinerary planning”, “virtual game-based design”, “cultural tourism features” and “tourism English” [17]. Hahn et al. initiated a user-centered design of a virtual reality heritage tourism system composed of three basic attributes: “VR environment”, “optimization” and “player interaction” [18]. Although various attributes related to VR tourism have been proposed based on their context, there is no consensus on the attributes of VR tourism. Moreover, it seems that prior research did not examine specific attributes from a holistic and systematic perspective because VR tourism attributes remain largely fragmented. 

In terms of more advanced approaches to VR tourism attributes, Philip and Hazlett proposed the hierarchical structure model called the PCP (pivotal, core, peripheral) attribute model, which can offer support in this area. In the model, pivotal attributes that focus mainly on output are considered attributes of the most intrinsic central level of quality of service. Pivotal attributes refer to the most decisive and core attributes in providing any product or service to consumers regardless of personal preference. In the case of QJK, fulfilment of travel needs and experience could be a pivotal attribute. Core attributes that users need to interact with to achieve the pivotal attribute are considered as the process and middle level. The core attribute, which encompasses the pivotal attribute, acts as a mediator to help realize the ultimate goal, a satisfying VR tourism experience. For this reason, ease of use, personal information protection, and security can be included in the category of core attribute. Peripheral attributes representing “completeness to the entire service encounter” or “roundness” are considered input and lowest level [19]. The peripheral attribute, in connection with the core and peripheral attributes, plays a role of making the product distinct from other types of products. In the case of VR tourism, tourists are expected to recognize the difference between VR travel and other previous tourism products mainly through the peripheral attribute. Specifically, in terms of VR tourism, interface design and operating system quality are expected to be included in the surrounding properties. Considering the characteristics of these three attributes, the PCP attribute model seems to contribute to developing a framework of VR attributes as a scientific body of knowledge [20,21,22].

The common underpinning paradigm of the PCP attribute model has received much attention from researchers in the field of marketing. The three-rank attribute model is always trimmed to a two-level attribute model when it is employed in practice, with peripheral (input) being the low level and core attribute (output) being the intrinsic ranking. For example, Skard et al. investigated consumers’ inferences about sustainable products with green core attributes and green peripheral attributes [23]. Wang et al. examined antecedents of brand experience in a historical and cultural theme park with the core and peripheral attributes [24]. However, limited research has investigated the PCP attribute of VR tourism. Built on the skeletal framework of the PCP attribute model and primary streams of literature about VR tourism, the current study develops a specific PCP attribute model for VR tourism to fill the gap. In the current study, VR tourism consists of pivotal attributes which indicate the output of VR tourism (e.g., VR tourism fits well with tourist’s travel needs), core attributes which refer to the process of VR tourism (e.g., VR tourism is easy to use) and peripheral attributes which represent the input of VR tourism (e.g., the user interface design is fascinating). The configuration of the PCP attributes is shown in Figure 1.

### 2.2. Presence and PCP Attribute of VR Tourism

Presence is the subjective experience of the VR environment, whilst users are physically in real world [6]. The term of presence, also known as telepresence, is widely accepted as a sense of “being there”, a psychological effect, in non-physical space [7,25]. Social psychology researchers and practitioners have noticed the significance of understanding presence and the relationship between presence and VR attributes. Presence is crucial for evaluating VR effectiveness [6,7,25,26]. To put it another way, when the level of presence experienced by a participant is low, the side effects may be produced. Nichols et al. addressed the important role of presence in VR and identified attributes (i.e., content and design, scene registration or update lags, head-mounted display optics and design, display and interaction) that may produce “sickness” [26]. Orth et al. postulated four informational attributes (i.e., mystery, complexity, legibility, and coherence) with construal level theory and examined how to achieve presence in virtual service environments [27].

The main discussion dominating the literature is that presence is characterized as transportation, a sort of subjective experience or sensation of “it is here” and “being there”. This sense of transportation is usually labeled using a two-dimension metaphor, arrival (being present in VR) and departure (not being present in VR) [7,28]. The discussion notes the critical dynamic process in which VR tourists continuously suppress input information that is incompatible with his or her desired VR experience and construct the mental model needed to experience presence [29]. The PCP attribute model proposed in the current paper is designed to understand this transportation process from input to output. In addition, the PCP model includes various determinants of presence which can be generally divided into external stimuli (VR environment delivery) and internal tendencies (user features) [6,30,31,32]. For example, richness, one of VR environment delivery, is reflected in an item of pivotal attribute (“VR tourism fits well with my travel needs”). Even if specific determinants are not included in the current item scale, they can be categorized as one of the PCP attributes. Consequently, the following hypotheses are proposed:

**Hypothesis 1** **(H1).**
*The peripheral attribute has a positive effect on presence in VR tourism.*


**Hypothesis 2** **(H2).**
*The core attribute has a positive effect on presence in VR tourism.*


**Hypothesis 3** **(H3).**
*The pivotal attribute has a positive effect on presence in VR tourism.*


### 2.3. Perceived Value and Presence during VR Touristic Experience

In general, value is an abstract and polysemic concept. The mainstream of the academic literature has been focusing on perceived value instead. Perceived value is a useful and critical construct for identifying tourist behavior in tourism industry. In many cases, perceived value has been regarded as a multidimensional concept, although sometimes as unidimensional one, with regards to overall value [33]. It is commonly understood from the consumers’ standpoint in the literature. In the early 1988, Zeithaml captured a widely accepted definition from four prior definitions. Perceived value is defined as “the consumer’s overall assessment of the utility of a product or service based on perceptions of what is received and what is given” [34]. It is built on the dual conception, “get-give” tradeoff [35]. Accordingly, VR tourists may trade off perceived benefits (e.g., convenience, utilitarian features, positive emotions) and perceived sacrifices (e.g., time, money, effort) [33]. The tradeoff conception conceives perceived value as a temporally dynamic process: pre-use, at the time of use and after use [36]. Although ubiquitous dimensions of perceived value are proposed in the literature, they echo the two underlying ones, functional value and emotional value [36,37,38,39,40]. Functional value refers to the rational and utilitarian value perceived by individuals. Emotional value is the feelings or affective states generated by a product or service [36,37].

The research on the consequences of presence in VR has converged on emotional response. For example, Yung et al. established a conceptual model comprising the consequences of presence in VR on emotional response by a critical review of presence research [6]. Gorini et al. discovered the similar findings when they evaluated the emotional response produced by VR [41]. Furthermore, presence is found to be crucial for improving perceived effectiveness and usability [42,43]. Brade et al. evaluated impact of presence on perceived usability using a mobile navigation task [43]. Likewise, Sun et al. demonstrated that presence is positively related to functional value in virtual environments [42]. Additionally, the extant literature has confirmed that tourist experience influences their perceptions of functional and emotional values. For example, Song et al. examined the impact of tourist experience on perceived value with temple stays [44]. According to the theory of presence, presence is a sort of subjective experience [6]. Above all, the hypotheses are suggested as follows:

**Hypothesis 4** **(H4).**
*Presence in VR tourism positively influences functional value.*


**Hypothesis 5** **(H5).**
*Presence in VR tourism positively influences emotional value.*


### 2.4. Satisfaction and Perceived Value

Customer satisfaction is defined as the measurement of how the actual experience generated by the service or product fulfills the customers’ expectations [45]. It is the central concept of marketing from which the term “tourist satisfaction” derived [46]. In the literature about tourism, the tourists’ overall satisfaction is usually in line with their levels of return visits to the destination, loyalty, and the retention of tourists [47]. Hence, managing tourist satisfaction is crucial for the successful development of the tourism industry. Accordingly, VR tourist satisfaction is of substantial importance for understanding the effectiveness and performance of VR tourism. In recent studies, research has been exploring the topic of satisfaction in VR. Hudson et al. investigated the moderating effect of immersion, interaction and social interaction in VR on users’ satisfaction [48]. Kim and Ko found that the effect of VR on flow experience, which will improve media user satisfaction, decreases as sport involvement increases [49]. Thus, satisfaction in VR has received much attention.

Besides, a range of tourism research has validated the relationship between satisfaction and perceived value, with an increasing number of studies reporting that tourist satisfaction is positively affected by perceived value. For example, Song et al. confirmed the clear relationship between tourist satisfaction and perceived value, showing that functional and emotional values influence tourist satisfaction [49]. Similarly, Wang et al. examined the positive impact of functional value and emotional value on consumers’ satisfaction level at a theme park [24]. However, few studies explore the effect of perceived value on satisfaction in VR tourism. To fill in the gap, we propose the following hypotheses:

**Hypothesis 6** **(H6).**
*Functional value has a positive effect on satisfaction in VR tourism.*


**Hypothesis 7** **(H7).**
*Emotional value has a positive effect on satisfaction in VR tourism.*


### 2.5. Subjective Well-Being and Satisfaction

The psychological study of subjective well-being has developed since Warner Wilson’s critical review [50]. Defined as people’s evaluation of their well-being, subjective well-being (SWB) is an essential element for improving positive physical and mental health and quality of life. [10,12,50]. In the context of tourism, SWB is, on one hand, the social outcome of tourism development: on the other hand, SWB is beneficial for tourism operators, policy makers, and tourists to promote understanding of the impacts of the tourism industry [51]. The fact that tourism contributes to tourist SWB has been confirmed in the literature. Meng et al. investigated the SWB of Chinese rural–urban migrants in the context of rural tourism, revealing that returning to rural destinations improves tourists’ SWB as they achieve an important lifetime goal via such experiences [52]. Through exploring the nature of tourists’ experiences, Knobloch et al. suggested understanding tourist consumption experiences beyond their momentary effects and considering a broader scope of well-being [53]. Thus, this empirical research rests on cognitive bases (e.g., the accomplishment of goals) and effects (hedonic balance) [51].

In the literature concerning the study of tourism, the research has recently started to focus on the link between tourist satisfaction and SWB. Saayman et al. investigated the impact of travel experience on tourists’ experience which further influence their SWB [54]. Similarly, Su et al. reported that overall customer satisfaction has a positive influence on SWB [51]. Nonetheless, such attempts are not observed in VR and VR tourism settings. Based on prior findings, the following hypotheses are formulated:

**Hypothesis 8** **(H8).**
*Satisfaction in VR tourism has a positive influence on SWB.*


## 3. Methodology

A thorough review of the literature concerning the related constructs and topics was undertaken before the original measurement items were developed. To ensure an appropriate questionnaire with good readability and effectiveness, two experts, and Mr. Ma, the chief executive of QJK, were asked to assess the content validity of the questionnaire and some obscure expressions in it were removed or modified. A 5-point Likert-type scale ranging from 1= “strongly disagree “to 5= “strongly agree” was applied to measure questionnaire items (see Appendix A). Built on the implication of PCP model and variables of VR tourism attributes in prior research, multiple items used to measure peripheral, core, and pivotal attributes were adopted [21,24,55]. In specific, the peripheral attribute was assessed using a 5-item scale including “operating system compatibility and applicability” and “interface design”. The core attribute was determined by a 4-item scale comprising “ease of use”, and “privacy and security”. The pivotal attribute was measured by a 5-item scale involving “fulfillment and advantage of VR tourism to users”. The 4-item scale of presence (e.g., “In the VR tourist world, I had a sense of being there”) was adapted from Bogicevic et al. and Schuemie et al. [25,29]. Based on suggestions from prior studies, two dimensions of perceived value (i.e., functional value and emotional value) were measured with four items respectively for each one (e.g., “The VR tour on QJK has a consistent level of quality” for functional value; “Using QJK for VR travel gives me a feeling of happiness” for emotional value) [33,36,44]. Satisfaction was operationalized with three items which were recommended by Lee et al. and Song et al. [40,44]. Finally, subjective well-being was assessed with four items, as suggested by Kim and Hall [8].

With the assistance of QJK, researchers contacted with VR tourists from the top four metropolises in China (i.e., Beijing, Shanghai, Guangzhou, and Shenzhen) via a Group Chat created by QJK. WeChat and QQ are the most prevalent social media platforms whose users are active. The Group Chats on WeChat and QQ were established as channels for QJK users’ to communicate with and give feedback to us. It was much easier to conduct the survey in the four cities which have a significant number of QJK’s users from various areas of China. Compared with an offline survey, it was more suitable to perform the research online in this study because the most of the users of QJK are active online and they registered with their real names. Based on this, an online anonymous survey was conducted among residents who have used QJK for VR travel during the COVID-19 pandemic (from February to November in 2020). The data were collected online from 19 November to 11 December 2020 by employing convenience sampling. The researchers sent friend requests to the potential respondents via WeChat or QQ in the first instance. Next, we described the purpose of survey, the time when responses were due, and compensation. After accepting our invitation on WeChat or QQ, each of respondents was asked to fill in the self-administered questionnaire online. 589 questionnaires were distributed and a total of 542 respondents completed the questionnaire. After excluding invalid questionnaires that were completed hastily or in repetitive response patterns, the remaining 490 completed questionnaires were finally used for the empirical analysis.

The data were analyzed using R and descriptive statistics was performed at first. Based on Anderson and Gerbing’s suggestions, the current study conducted structural equation modelling (SEM) with a two-step approach [56]. In order to ensure internal consistency together with construct validity and reliability, confirmatory factor analysis was firstly implemented to examine the measurement model for all variables. Moreover, SEM was performed to examine the proposed research model and hypotheses. Figure 2 is the proposed conceptual model.

## 4. Results

### 4.1. Descriptive Statistics

The complete respondent demographic characteristics are provided in Table 1. As shown in Table 1, the number of females (51.6%) was slightly higher than that of males (48.4%). Among the 490 respondents, 31.2% were single and 59.2% were married. A wide range of occupations were present, including technicians and professionals (20.4%), businessmen and self-employed (23%), service workers (5.7%), office workers (8.4%), officials (10.4%), students (10%), freelancers (13.5%), and retired people (4.9), with the level of education ranging from less than high school (11.8%), to a postgraduate degree (18.2%). The majority of reported monthly incomes were more than CNY 5000. In terms of residence, 21.2% of respondents were in Beijing, 24.9% were in Shanghai, 27.4% were in Guangzhou, and 26.5% were in Shenzhen. The age group of 20–29 years old was dominant, representing 37.1%, followed by age groups of 30–39 years old (29.2%) and 40–49 years old (21.6%).

### 4.2. Measurement Model

Generally, two approaches are used to assess structural equations, maximum likelihood (ML) and robust methods. The commonly used ML estimation is used when the data follow the assumption of a multivariate normal distribution. If the data do not meet the criteria for a multivariate normal distribution, the study results provided through ML are considered unreliable [57]. In this case, another approach like robust estimation should be performed. To test the multivariate normal distribution assumption, Mardia’s standardization coefficient is used. If the value exceeds 5, the collected data are considered not to satisfy the assumption of multivariate normal distribution. In this study, the MLM (maximum likelihood estimation with robust standard errors and a Satorra–Bentler scaled test statistic) estimator, which is one of the powerful methods, was used because Mardia’s standardization coefficient (66.032) in this study exceeded the cutoff value of 5 [58].

Hair et al. suggested that normed S-B χ^2^ below 3 is associated with a good model fit if sample size is less than 750. Values of 0.9 or greater show good model fit for the Comparative Fit Index (CFI), Normed Fit Index (NFI) and Non-Normed Fit Index (NNFI). For the Root Mean Square Error of Approximation (RMSEA), a cut-off criterion is needed [59]. As presented in Table 2, the overall fit of the measurement model is satisfactory: S-B χ^2^ (df) = 739.268 (467), Normed S-B χ^2^ = 1.583, CFI (Comparative Fit Index) = 0.975, NFI (Normed Fit Index) = 0.936, NNFI (Non-Normed Fit Index) = 0.972, RMSEA = 0.034.

The Cronbach’s alpha of the latent variables varied from 0.863 to 0.939, indicating the acceptable reliability of the measurement model. The standardized factor loadings of the items ranged from 0.772 to 0.907, which were all statically significant (*p* < 0.001) and exceeded the cut-off point of 0.5. The values of average variance extracted (AVE) were all greater than the recommended value of 0.5, varying from 0.675 to 0.782. Composite reliability (CR) for all variables ranged from 0.862 to 0.940, which exceeded the critical value of 0.7 [60]. In addition, all AVE values were greater than the values of squared correlations among latent constructs [61]. Therefore, the convergent and discriminant validity was confirmed [59].

### 4.3. Structural Model

As illustrated in Figure 3, the overall fit of the structural model is satisfactory: S-B χ^2^ (df) = 886.124, Normed S-B χ^2^ = 1.831, CFI = 0.964, NFI = 0.923, NNFI = 0.960, RMSEA = 0.041. Based on the cut-off values suggested in the prior discussion, the results demonstrate that the structural model’s fit is satisfactory.

As for Hypothesis 1, which predicted a positive relationship between peripheral attribute and presence was supported (β _PEA→PRE_ = 0.199, t = 4.597, *p* < 0.001). The hypothesized positive relationship between core attribute and presence (H2) was accepted (β _CA→PRE_ = 0.315, t = 7.330, *p* < 0.001). As presumed by Hypothesis 3, the pivotal attribute had a positive effect on presence in VR tourism (β _PIA→PRE_ = 0.439, t = 11.201, *p* < 0.001). It presented that presence positively influenced functional value (β _PRE→FV_ = 0.696, t = 22.997, *p* < 0.001) and emotional value (β _PRE→FV_ = 0.684, t = 21.012, *p* < 0.001). Function value (β _FV→SAT_ = 0.354, t = 11.054, *p* < 0.001) and emotional value (β _EV→SAT_ = 0.576, t = 20.201, *p* < 0.001) were each found to have a positive impact on satisfaction. Finally, satisfaction positively influenced subjective well-being (β _SAT→SWB_ = 0.783, t = 28.622, *p* < 0.001). The predicted relationships, coefficients, t-values and results of hypothesis test are shown in Table 3.

## 5. Discussion and Limitation

### 5.1. Discussion

This study established and tested an integrated model with constructs of VR tourism attribute, presence during VR tourism experience, perceived value of VR tourism, satisfaction with VR tourism experience and VR tourists’ subjective well-being. Firstly, built on Philip and Hazlett’s proposed the hierarchical structure model, the PCP attributes of VR tourism was developed [20,21,22]. Our results indicate that VR attribute positively influences presence during VR tourism experience. Specifically, the pivotal attribute has the strongest impact on presence among the other two sorts of attributes, with the core attribute being the second strongest and the peripheral attribute ranking the last. This finding accords with Philip and Hazlett’s previous judgements, which considered the pivotal attribute to be the center-level, the core attribute to be the median and pivotal to be the exterior [21]. In practice, the three levels are usually reduced to two, namely the peripheral and core attribute. Similarly, our findings are also consistent with results of prior research that reported the stronger influence of the core attribute on tourist experience [24,40]. There is no consensus on the attributes of VR tourism in the related literature. Elements of VR tourism products or services determining presence during VR experiences were also revealed. To put it another way, VR tourists place more weight on core attributes (e.g., “VR product is easy to use”) and pivotal attributes (e.g., “VR product fits travel needs”) than peripheral attribute (e.g., “user interface design”), while peripheral attribute is not dispensable. Secondly, presence during VR tourism experience is demonstrated to positively affect two constructs of perceived value to almost the same degree. Presence denotes a psychological effect of “being there” in virtual space. This shows the dynamic process that VR tourists subjectively select their expected information during VR travel to receive the desired emotional and functional values which have equal importance to them. The impact of presence during VR experience on emotional response, perceived effectiveness and usability has been evaluated in the extant literature [41,42,43,44]. However, this is the first study to explore the relationship between presence and perceived value in the domain of VR tourism. Filling this gap is important not merely due to extending the VR tourism literature by covering a relationship that tends to be ignored, but also because perceived value is beneficial to predicting the effectiveness and performance of VR tourism [34,35]. Thirdly, a significant positive effect of perceived value on satisfaction confirms that the extent to which VR tourists’ emotional reactions and VR tourism products or services influence their overall evaluation of VR tourism. Consistent with clearly certified correlations in previous research, emotional value was found to be more significant than functional value in influencing VR tourists’ satisfaction [24,40]. This result suggests that focusing on the emotional design of VR tourism tends to make it easier to foster satisfaction in VR tourism and further improve residents’ well-being. Finally, satisfaction positively leads to subjective well-being, as explained by 61.3% of the sample. It is important to note here that residents’ subjective well-being will be improved if VR tourism can be served as their satisfactory means of leisure activities. Therefore, the critical role of VR tourism in improving residents’ well-being has been proved.

### 5.2. Limitations and Future Research

Like all studies, the present study has some limitations that warrant consideration in future research. These investigations were only performed in the largest metropolises in China (Beijing, Shanghai, Guangzhou and Shenzhen) and the future research should consider a comparative study in other geographical areas. Further, despite the PCP attribute model provides a framework for exploring attributes of VR tourism, the importance and performance of each specific attribute are not assessed. Evaluating each specific attribute may help VR tourism producers understand their products and VR tourist feedback. Importance-performance matrix should be employed for future investigation. Besides, the survey was conducted among respondents who have used QJK during the pandemic, while users’ acceptance of VR tourism was not analyzed. Future researchers could explain VR travel intentions and subsequent behaviors with models like UTAUT model (the unified theory of acceptance and use of technology model). Future researcher should consider to perform psychological experiments onsite among those who have no experience of VR travel and thereby recommend effective marketing strategies for tourism operators who are eager to see the recovery from recession of tourism industry. Likewise, the same patterns of experiments should also be carried out among international tourists after the pandemic. VR tourism offers opportunities for marketers to communicate their tourism products to potential visitors and enhances mutual understanding between tourists from different countries as they trust better their peers than marketers [6,62].

## 6. Conclusions

In the landscape of the COVID-19 pandemic, the tourism industry has been struggling due to the recession and the postponement of trips to tourist destinations. Residents’ well-being, to some extent, has been impacted directly or indirectly due to the spread of the coronavirus. VR tourism, as a form of leisure activity in daily life, provides an effective coping strategy to enhance residents’ levels of well-being and destination recovery. In such a context, an investigation was conducted among residents who used QJK, the largest tourism-related VR website in China, during the pandemic. The aim of this paper is to provide empirical evidence to prove the effectiveness of VR tourism in promoting residents’ subjective well-being. Hence, an integrated model with the constructs of PCP attributes, presence, perceived value, satisfaction, and subjective well-being was proposed and tested. The results indicate that the PCP attributes of VR tourism have a positive effect on presence during VR experience, which positively influences perceived value. The value of VR tourism as perceived by VR tourists results in their satisfaction. Eventually, residents’ subjective well-being is improved due to their satisfaction with VR tourism. Based on our findings, suggestions for policy makers, residents and tourism operators are offered as follows:

Policy makers should make constructive use of leisure activities associated with a high level of residents’ well-being such as VR tourism while they are striving for economic development and social stability. The local government may collect applicable information and data via big data about VR tourism to make the city more livable, workable and sustainable. In particular, VR tourism can facilitate the disabled with access to destinations in VR, which will, to some degree, contribute to realizing government’s goal for social equity. Meanwhile, we recommend that local residents accept and enjoy virtual travel as VR tourism has great potential to improve their well-being, provides all sorts of travel related information to help them pursue their interests, saves time and money, and allows them to connect with friends while traveling in VR. For destination suppliers, VR tourism is capable of retaining the demands of future tourists and thereby provides practical solutions for destination recovery after the pandemic because VR tourism is associated with real visitation and intention to travel [6,12]. Therefore, it is suggested that destination suppliers cooperate with VR tourism developers. VR tourists favor the VR tourism products and services that present them with a high degree of presence. Thus, developers should highlight core and pivotal attributes when they design VR projects. Subsequently, VR tourists could achieve intensive presence and perceived positive values that will result in their satisfaction with the destination and VR tourism. All in all, VR tourism is supposed to be applied in multiple sectors for various purposes.

## Figures and Tables

**Figure 1 ijerph-18-01019-f001:**
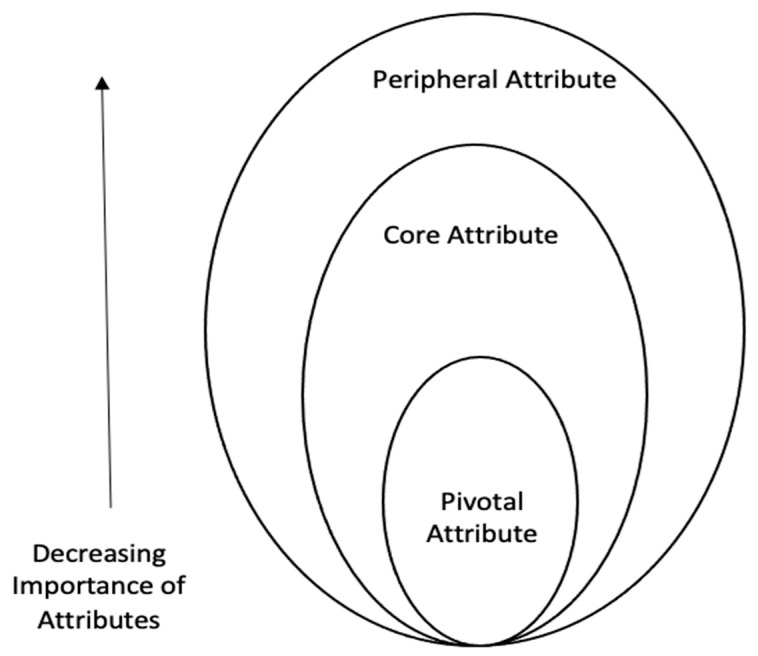
Conceptual framework for PCP (pivotal, core, peripheral) model of VR (Virtual Reality) tourism attribute.

**Figure 2 ijerph-18-01019-f002:**
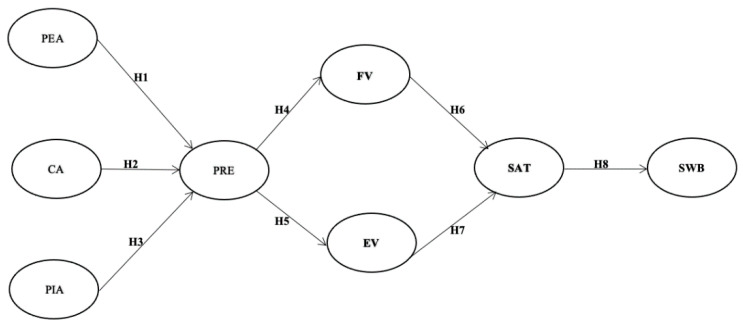
Proposed conceptual model Notes 1: PEA = peripheral attribute; CA = core attribute; PIA = pivotal attribute; PRE = presence; FV = functional value; EV = emotional value; SAT = satisfaction; SWB = subjective well-being.

**Figure 3 ijerph-18-01019-f003:**
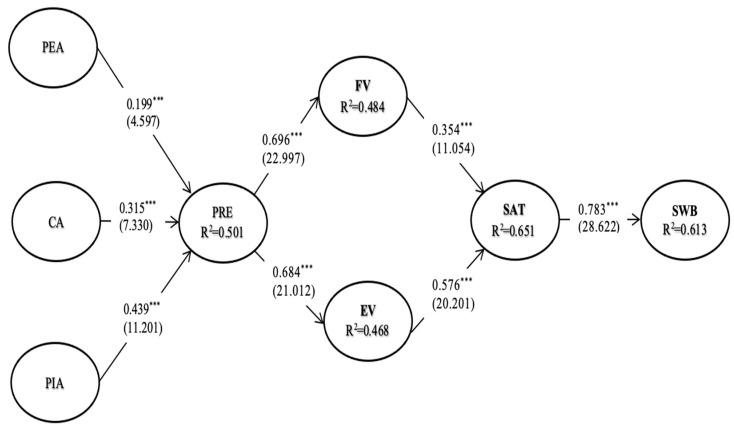
Structural model results. Note 1: *** *p* < 0.001. Note 2: Values not in parentheses are standardized parameter estimate; values in parentheses are t values. Note 3: PEA = peripheral attribute; CA = core attribute; PIA = pivotal attribute; PRE = presence; FV = functional value; EV = emotional value; SAT = satisfaction; SWB = subjective well-being. Note 4: S-B χ^2^ (df) = 886.124; Normed S-B χ^2^ = 1.831; CFI = 0.964; NFI = 0.923; NNFI = 0.960; RMSEA = 0.041.

**Table 1 ijerph-18-01019-t001:** Respondents’ demographic characteristics (*n* = 490).

Characteristic	N (%)	Characteristic	N (%)
Gender		Marital status	
Male	237 (48.4)	Single	153 (31.2)
Female	253 (51.6)	Married	290 (59.2)
		Others	47 (9.6)
Education level		Monthly income level ^a^	
Less than high school	58 (11.8)	Less than 3000	49 (10)
Three-year college	136 (27.8)	3000–4999	90 (18.4)
Four-year university	207 (42.2)	5000–6999	209 (42.6)
Graduate school	89 (18.2)	7000–8999	94 (19.2)
		9000 or more	48 (9.8)
Occupation		Residence	
Technicians/Professionals	100 (20.4)	Beijing	104 (21.2)
Businessmen/Self-employed	113 (23)	Shanghai	122 (24.9)
Service workers	28 (5.7)	Guangzhou	134 (27.4)
Office workers	41 (8.4)	Shenzhen	130 (26.5)
Official	51 (10.4)		
Students	49 (10)	Age	
Freelancers	66 (13.5)	Less than 20	18 (3.7)
Retire	24 (4.9)	20–29	182 (37.1)
Others	18 (3.7)	30–39	143 (29.2)
		40–49	106 (21.6)
		50–59	32 (6.5)
		Over 60	9 (1.9)

^a^ USD 1 is equivalent to CNY 6.55.

**Table 2 ijerph-18-01019-t002:** Results of measurement model.

Constructs	PEA	CA	PIA	PRE	FV	EV	SAT	SWB	Items	Standardized Factor Loading
Peripheral attribute (PEA)	0.758	0.181 (0.425)	0.133 (0.364)	0.240 (0.490)	0.114 (0.338)	0.139 (0.373)	0.112 (0.335)	0.092 (0.303)	PEA 1 PEA 2 PEA 3 PEA 4 PEA 5	0.841 0.896 0.907 0.881 0.826
Core attribute (CA)	0.032	0.691	0.035 (0.186)	0.229 (0.479)	0.104 (0.323)	0.139 (0.373)	0.101 (0.318)	0.077 (0.277)	CA 1 CA 2 CA 3 CA 4	0.772 0.860 0.851 0.839
Pivotal attribute (PIA)	0.035	0.030	0.737	0.324 (0.569)	0.146 (0.382)	0.168 (0.410)	0.082 (0.286)	0.100 (0.316)	PIA 1 PIA 2 PIA 3 PIA 4 PIA 5	0.850 0.837 0.864 0.893 0.848
Presence (PRE)	0.040	0.035	0.040	0.782	0.458 (0.676)	0.434 (0.659)	0.362 (0.602)	0.266 (0.516)	PRE 1 PRE 2 PRE 3 PRE 4	0.852 0.895 0.907 0.883
Functional value (FV)	0.033	0.028	0.034	0.043	0.709	0.497 (0.705)	0.430 (0.655)	0.520 (0.721)	FV 1 FV 2 FV 3 FV 4	0.821 0.862 0.823 0.862
Emotional value (EV)	0.034	0.031	0.036	0.043	0.039	0.703	0.544 (0.738) *	0.530 (0.728)	EV 1 EV 2 EV 3 EV 4	0.775 0.864 0.864 0.847
Satisfaction (SAT)	0.033	0.027	0.032	0.040	0.038	0.054	0.675	0.532 (0.729)	SAT 1 SAT 2 SAT 3	0.791 0.856 0.817
Subjective well-being (SWB)	0.033	0.028	0.034	0.038	0.038	0.037	0.054	0.701	SWB 1 SWB 2 SWB 3 SWB 4	0.799 0.860 0.870 0.818
CR	0.940	0.899	0.933	0.935	0.907	0.904	0.862	0.903	Model fit S-B χ^2^(df): 739.268 (467) Normed S-B χ^2^: 1.583 CFI: 0.975 NFI: 0.936 NNFI: 0.972 RMSEA: 0.034	
Cronbach alpha	0.939	0.898	0.933	0.934	0.906	0.901	0.863	0.903		

*: Highest correlation between pairs of construct; The values of AVE highlighted in shade are along the diagonal. Squared correlations among latent constructs are above the diagonal. Correlations among latent constructs are within parentheses. Standard errors among latent constructs are below the diagonal. Mardia’s normalized coefficient: 66.032. All standardized factor loadings are significant at *p* < 0.001.

**Table 3 ijerph-18-01019-t003:** Standardized parameter estimates of structural model.

	Hypotheses	Coefficients	t-Value	Test of Hypotheses
H1	PEA→PRE	0.199	4.597	Accepted
H2	CA→PRE	0.315	7.330	Accepted
H3	PIA→PRE	0.439	11.201	Accepted
H4	PRE→FV	0.696	22.997	Accepted
H5	PRE→EV	0.684	21.012	Accepted
H6	FV→SAT	0.354	11.054	Accepted
H7	EV→SAT	0.576	20.201	Accepted
H8	SAT→SWB	0.783	28.622	Accepted

## Data Availability

The dataset used in this research are available upon request from the corresponding author. The data are not publicly available due to restrictions i.e., privacy or ethical.

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
