# Peer review of "A Study on the Causal Process of Virtual Reality Tourism and Its Attributes in Terms of Their Effects on Subjective Well-Being during COVID-19"

_ijerph, 2021, doi:10.3390/ijerph18031019_

Round 1
Reviewer 1 Report
I congratulate the authors for the concepts of research. The subject is well described and the exposition is clear even to a reader unfamiliar with the topic. The subject is very up-to-date, maybe not yet in Europe, but certainly the "migration" to virtual tourism because of COVID-19 will be huge. The paper represents a very good preparation of the authors for the project, with an in-depth analysis of the latest literature. The bibliography is very current and agrees with the topic of study.
I have only two minor comments:
-
I do not understand the purpose of using italic in 2.1 section i.e. accurate, crucial, prevalent
- It's a little bit of an eye-catching to present the purpose of the study in the present time, maybe it's just my habit, but it seems to me that the purpose of the study should be planned before it begins and thus determined in the past tense.
Reviewer 2 Report
1. We need a research model. A research model that can grasp the purpose and design of the paper at a glance is drawn in '3. Please include it in'Methodology'.
2. QJK should be added to explain whether it is a reliable website and whether data collection is meaningful.
3. Literature review related to the integration of the PCP model's tourism field or service field is needed more intensively. This is an important variable, but there is not much explanation for this. Add a review of the literature for that model.
4. There is a need for logical support that can prove the limitations and reliability of conducting online surveys only. Add no difference to offline surveys and a methodology to build trust in your data.
5. The results analyzed using R studio are not included. Please add the report according to the suggested research method.
Reviewer 3 Report
Please avoid use of acronymous in the abstract
Website of Chinas greatest portal of VR should be explained in more details: website, languages available, some statistics
Recruitment: How did you recruited people who had seen at least once this portal
Suggestion: for ease of readiness sum up in a graph the main components of VR frameworks of previous scholars (eg. Argyriou et al.’s model, PCP’s model of Philip and Hazlett)
Reviewer 4 Report
This is an interesting paper on a very recent theme and in which studies are still at an early stage to ensure the certainty of viability as a solution, for the recovery of tourism worldwide in the post-pandemic era by covid-19.
I therefore find the results obtained to be inconsistent. The authors themselves are aware of the limitations of their research and of the results achieved!
As this is a case study, the title should be changed to the extent that the survey was only carried out for residents who used QJK for VR travel during the Covid-19 pandemic.
The number of responses treated is also not very significant compared to China's total population. Even in relation to the four main metropolises of China (Beijing, Shanghai, Guangzhou and Shenzhen), as mentioned, is a very low number.
A comparative study of the methodologies used in the study in other geographical areas and naturally related to the same theme would have been important.
In terms of the construction of the article I would have liked to have seen a review of the literature associated with a substantial conceptual definition, as well as the construction of the conceptual scheme that does not exist.
The literature review divided by subtitles is not very common.
This is an interesting article, but very difficult to read given the constant presence of acronyms, so the perception of the contents of the text also does not facilitate understanding, being little accessible.
Round 2
Reviewer 2 Report
You have really struggled to work on the revised version. Please correct grammatical errors and typos to complete the final version. A big applause for your efforts. Thank you.